# Inhibition of *Staphylococcus aureus* LC 554891 by *Moringa oleifera* Seed Extract either Singly or in Combination with Antibiotics

**DOI:** 10.3390/molecules25194583

**Published:** 2020-10-07

**Authors:** Gamal Enan, Abdul-Raouf Al-Mohammadi, Samir Mahgoub, Seham Abdel-Shafi, Eman Askar, Mohamed F. Ghaly, Mohamed A. Taha, Nashwa El-Gazzar

**Affiliations:** 1Department of Botany and Microbiology, Faculty of Science, Zagazig University, Zagazig 44519, Egypt; emanaskar91@gmail.com (E.A.); mfarouk2005@yahoo.com (M.F.G.); dr.taha_virus@yahoo.com (M.A.T.); mora_sola1212@yahoo.com (N.E.-G.); 2Department of Sciences, King Khalid Military Academy, P.O. Box 22140, Riyadh 11495, Saudi Arabia; almohammadi26@hotmail.com; 3Department of Agricultural Microbiology, Faculty of Agriculture, Zagazig University, Zagazig 44511, Egypt; mahgoubsamir@gmail.com

**Keywords:** moringa, *Staphylococcus aureus*, virulence factors, MSE, GC-MS analysis

## Abstract

Bacterial outbreaks caused by *Staphylococcus aureus* (*S. aureus*) are interesting due to the existence of multidrug resistant (MDR) isolates. Therefore, there is a need to develop novel ways to control such MDR *S. aureus*. In this study, some natural agents such as honey bee (HB), extracts of either *Moringa oleifera* seeds (MSE), or leaves (MLE) and essential oils of garlic, clove, and moringa were studied for their inhibitory activity against this *S. aureus* pathogen. About 100 food samples including beef luncheon (*n* = 25), potato chips (*n* = 50), and corn flakes (*n* = 25) were investigated for possible pollution with the *S. aureus* bacteria. The isolated bacteria suspected to belong *S. aureus* that grew well onto Baird–Parker agar (Oxoid) and shiny halo zones and positive coagulase reaction were selected and identified by API-Kits; all of them that were approved belong to *S. aureus* (18 strains). The sensitivity of the obtained 18 *S. aureus* bacterial strains to 12 antibiotics were evaluated; all of them were resistant to ofloxacin; however, other antibiotics tested showed variable results. Interestingly, the *S. aureus* No. B3 isolated from beef luncheon was resistant to 10 antibiotics out of 12 ones tested. Multiple antibiotic resistance index (MAR) of this *S. aureus* strain was about 83.3%. Therefore, its identification was confirmed by sequencing of a 16S rRNA gene which approved a successful biochemical identification carried out by API Kits and such strain was designated *S. aureus* LC 554891. The genome of such strain appeared to contain *mecA* gene encoding methicillin resistance; it was found to contain *hla, hlb, tsst-1,* and *finbA* that encode α-blood hemolysis, β-blood hemolysis, toxic shock syndrome gene, and fibrinogen-binding protein gene, respectively. In addition, the virulence factors viz. *sea; seb; sec* encoding enterotoxins were detected in the DNA extracted from *S. aureus* B3 strain. Aqueous extract of *Moringa oleifera* seeds (MSE) showed inhibitory activity against *S. aureus* LC 554891 better than that obtained by tetracycline, essential oils or HB. Minimum inhibitory concentration (MIC) of MSE was 20µg/mL. Instrumental analysis of MSE showed 14 bioactive chemical compounds. Combinations of both MSE and tetracycline showed distinctive inhibitory activity against *S. aureus* LC 554891 than that obtained by either tetracycline or MSE singly.

## 1. Introduction

*Staphylococcus aureus* is one of the most opportunistic pathogens associated with hospital and community acquired infections [1,2]. It is a common causal pathogen of skin abscesses, pharyngitis, sinusitis, meningitis, pneumonia, osteomyelitis, endocarditis, toxic shock syndrome, sepsis, and wound infections following surgery [3]. It is also responsible for food poisoning illness as it is capable of producing several virulence factors such as enterotoxins, adhesins, hemolysins, invasins, superantigens, and surface factors that inhibit its phagocytic engulfment [4].

*S. aureus* is a Gram-positive, coccoid-shaped, facultatively anaerobic and catalase positive bacterium. It forms yellow colonies on routine agar medium and forms black colonies with halo zones after its growth on its specific medium Baird–Parker agar due to telluride reduction of the medium and both lecithinase and coagulase activity. *S. aureus* is non-motile, non-spore former, ferments glucose, and produces lactic acid; it shows both α- and β- blood hemolysis capability and is characterized by positive coagulase reaction [1]. 

Food handlers carrying enterotoxin-producing *S. aureus* in their noses or on their hands are regarded as the main source of food contamination via manual contact or through respiratory secretions. *S. aureus* is the most abundant skin colonizing bacterium and the most important causes of mucosal infections and community-associated skin infections [3]. The contamination with *S. aureus* is due to improper handling of ready-to-eat foods which allow growth of *S. aureus* and production of enterotoxin (s) [3,4].

Recent studies have shown different levels of percentage values of incidence of *S. aureus* in foods [5]. It reached 4% in raw pasteurized milk, 40% in beef luncheon, 20–40% in corn flakes, and almost 12.5% in different Chinese foods [6,7]. Therefore *S. aureus* is a world health problem and there is a need to continue research to find out novel ways for its inhibitory therapy either in vivo or in foods.

The severity of *S. aureus* infection is currently being increased due to emergence of multidrug resistant strains of *S. aureus* which are becoming endemic worldwide and are spreading into the community at large [8]. Vancomycin intermediate *S. aureus* (VISA) and vancomycin resistant *S. aureus* (VRSA) were isolated from different medicinal samples [8]. Therefore, inhibition of *S. aureus* by other alternatives is mandatory. In this regard, plant extracts are used nowadays, *Moringa oleifera* extracts of either leaves or seeds inhibited different pathogenic bacteria in vitro, HB and garlic extracts are used also as an antibacterial agent [9,10,11]. This is to concur with the international interest to inhibit the multidrug resistant bacteria, in general, and *S. aureus* in particular.

*M*. *oleifera* is a medicinal plant, a rich source of bioactive compounds and is used in the treatment of certain diseases. *M*. *oleifera* inhibit Gram-positive and Gram-negative bacteria including *Staphylococcus aureus Bacillus cereus*, *Escherichia coli*, *Salmonella enteritidis*, and *Pseudomonas aeruginosa*. Its extracts contain alkaloids, steroids, triterpenes, flavinoid, polyphenols with antibacterial activities. Moreover, *M*. *oleifera* has several peptides with antimicrobial activities [12]. *M. oleifera* seed extract exerts its protective effect by decreasing liver lipid peroxides, antihypertensive compounds thiocarbamate and isothiocyanate glycosids which have been isolated from the acetate phase of its ethanolic extract [12,13].

Numerous studies have been published on the antibacterial activities of honey showing its biological activities [14]. It is used as antibacterial agent against antibiotic-resistant bacteria [15]. Antibiotic susceptible and resistant isolates of *S. aureus, S. epidermidis, Enterococcus faecium, E. coli, Strept. pyogenes, P. aeruginosa, Enterobacter cloacae,* and *K. oxytoca* were killed within 24 h by 10–40% (*v*/*v*) honey [16].

The aim of the present work was to (i) assess the possible pollution of some ready-to-eat Egyptian foods *S. aureus* bacteria, (ii) study the antibiotic sensitivity of the obtained bacteria, and (iii) study the inhibition of MDR *S. aureus* LC 554891 strains by natural inhibitory agents either singly or in combination with antibiotics.

## 2. Results

### 2.1. Isolation and Identification of Presumptive Staphylococcus aureus Strains from some Egyptian Foods

One hundred food samples including beef luncheon (*n* = 25), potato chips (*n* = 50) and corn flakes (*n* = 25) were tested for existence of the presumptive *S. aureus* colonies. About 30 food samples showed total Staphylococci counts, 18 samples (18% of the total tested) of them showed SAC as food pollutants. Beef luncheon and chips showed presumptive *S. aureus* counts above the allowed standards (>5 × 10^3^ CFU/g) by about 45.45% and 28.57%, respectively, within the positive samples that showed bacterial counts (Appendix A). The all 18 presumptive *S. aureus* isolates obtained were Gram positive and catalase positive coccid cells. They were identified by API-Kits according to the manufacturer’s instructions (Biomerereux, Montaliea, France). Those 18 bacterial isolates were identified as belonging to *S. aureus* bacterium (Appendix A).

### 2.2. Antibiotic Sensitivity Test

The antibiotic susceptibility of the *S. aureus* strains (*n* = 18) was studied. Results are given in Table 1. The MAR index of *S. aureus* No. B3 isolated from beef luncheon was of about 83.3%; and was of about 58.33% for the strain B8, B24, and Ch 41. The antibiotics tested could be arranged in the following descending manner according to their inability to inhibited the indicator bacteria: ofloxacin (100%) > tetracycline (84.2%) > oxacillin and doxycycline (52.63%) > ampicillin (47.36%) > neomycin and amoxicillin (42.10%) > ciprofloxacine, clindamycin and penicillin (36.84%) > spiramycin (26.31%) > methicillin (21.05 %). Hence, the *S. aureus* No. B3 strain was resistant to 10 antibiotics tested including the antibiotic methicillin. This showed that this strain was preliminary approved to be MDR strain. For more confirmation of its biochemical identification, this B3 strain showed positive reactions regarding coagulase, α- and β- blood hemolysis. This strain was selected to be identified by 16S rRNA and tested for its virulence at the molecular level.

### 2.3. Molecular Identification of the B3 Strain by Sequencing of the 16S rRNA Gene

Studies were further conducted on this selected B3 strain to confirm its identification by fingerprinting of the sequence of its 16S rRNA gene. To carry out this experiment, DNA was extracted from this isolate and PCR of the 16S rRNA gene was carried out. The amplified PCR product was electrophoresed and showed a DNA band of a molecular size of about 1400 bp “Figure 1”. This DNA band was cut by the gene purification kit sequenced, and this RNA sequencing “Appendix A” was submitted to the Gene Bank under accession number LC554891 to be compared with the stored ones using Basic Local Alignment Search Tool Programme. Similarity of the 16S rRNA gene sequence of the *S. aureus* B3 strain was 99.5% for the *S. aureus* category. For the relevant phylogenetic tree of the identified B3 strain, Figure 2 demonstrated that the B3 strain belongs to the *S. aureus* category; this B3 strain was designated *S. aureus* LC 554891.

### 2.4. Detection of Virulence Factors within an S. aureus LC 554891 Genome

The virulence genes within the *S. aureus* LC 554891 genome such as staphylococcal enterotoxins (*sea, seb, sec*), toxic shock syndrome toxin gene 1 (*tsst-1*), and fibrinogen-binding protein, (*fnbA*), were detected by primers given in Materials and Methods. DNA preparation was mixed with the primers used and the PCR rounds were done. The PCR products were electrophoresed using agarose gel (0.8 %). Results have showed that DNA bands of molecular sizes of about 120, 478, 257, 350, and 1362 bp were showed, indicating the virulence factors: *sea*, *seb, sec, tsst-1* and *fnbA*, respectively, Figure 3.

### 2.5. Genetic Linkage of the mecA Gene and Hemolysin Toxin hla and hlb Genes within the S. aureus LC 554891 Strain

The *mec A* gene; *hla*; *hlb* genes encoding resistance of *S. aureus* LC 554891 to methicillin, α blood hemolysis, β blood hemolysis, respectively, were detected by a PCR technique using specific primers given in Materials and Methods. The PCR products were electrophoresed via agarose gel and a DNA band of about 533 bp, indicating existence of *mecA* gene within the *S. aureus* LC 554891 genome, was showed in Figure 4A. In addition, DNA bands of molecular sizes of about 306 bp; 833 bp were found, indicating the *hla* gene, *hlb* gene which encode α- and β- blood hemolysis, respectively (Figure 4A,B). All 18 *S. aureus* strains obtained showed positive results regarding coagulase reaction, α and β-blood hemolysis.

### 2.6. Antibacterial Activity of some Essential Oils against S. aureus LC 554891 by the Disc Diffusion Assay

The effect of different natural agents that were available as essential oils of garlic, moringa, and clove were given in Table 2 and Appendix A. The antibiotic tetracycline (10 µg/mL); *S. aureus* strain ATCC 6538 were used as an antibiotic control, control strain, respectively. The essential oils of three plants inhibited *S. aureus* LC 554891 at both concentrations used (0.25% and 0.5%). The inhibitory activity against *S. aureus* LC 554891 was arranged almost in the following descending order: moringa oil ˃ garlic oil ˃ clove oil. Thus, *Moringa oleifera* was a subject for future investigation.

### 2.7. The Antibacterial Activity of M. oleifera Leaves Extract (MLE), M. oleifera Seeds Extract (MSE) and Honey Bee (HB) either Singly or in Combinations of MSE Plus Tetracycline

The inhibitory effect of MLE, MSE and HB against *S. aureus* LC 554891 is given in (Table 3) and Appendix A. MSE showed the higher inhibitory activity as inhibition zone diameters (IZDs) against *S. aureus* LC 554891 reached 47–50 mm, while inhibitory activity of either MLE or BH was rather low as IZDs reached up to 15 mm of MLE ethanol extract (EE) and up to 34 mm of 100% HB (Table 3). Water showed the best extraction solvent than either ethanol or methanol as IZDs obtained against *S. aureus* by MSE water extract (WE) reached 34–50 mm and were up to 0–10 mm; 0–18 mm by MSE (EE); MSE methanol extract (ME), respectively. Since MSE showed the best inhibitory activity against *S. aureus* LC 554891, its MIC value was carried out by a disc diffusion assay using Muller Hinton agar. As given in Figure 5, the MIC value of MSE was 20 µg/mL. The inhibition of the MDR *S. aureus* LC 554891 bacterium by MSE in this investigation is very important, and its inhibition by a natural agent either singly or in combination with antibiotics is very promising. Hence, results were further examined to check the effect of combinations of the antibiotic tetracycline with MSE. The data given in Figure 6 showed that mixing of the MIC value of MSE with 10 µg tetracycline inhibited *S. aureus* LC 554891 distinctively. Doubling the values of MIC of MSE with the same concentration of the antibiotic tetracycline showed wider inhibition of *S. aureus* LC 554891 that obtained by either MSE or tetracycline only. 

### 2.8. Instrumental Analysis of MSE by GC- MS

In the current study, MSE was subjected to GC-MS analysis to detect its bioactive compounds. GC-MS analysis showed about 16 principal peaks which are corresponding to more bioactive 14 compounds. The results obtained Table 4 and Figure 7 representing the name and classes, in addition to molecular formula and molecular weight, for the 14 organic chemical categories. The main compounds in the MSE are esters: Methyl 3-[3,5-di-tert-butyl-4-hdroxy phenyl] propionate and Bis [2-ethylhexyl] phlthalate;spiroketone:7,9-Di-tert-butyl-1-oxaspiro [4,5] deca-6,9-diene-2,8-diene; ketone: 6-Iodoacetoveratroneand 2-Alyl-5-t-butyl hydroquinone; heterocyclic compounds:1-Methyl-2-cyano-3-ethyl-4-pivaloyl-2-piperidine;fatty-ester:ethylhexadecanoate, Hydroxy ethyl myristate, 2-hydroxy ethyl palmitate and 1-(Hydroxymethyl)-1,2-etheraneelyl ester dibasic fatty acid: octadecanoic acid; polynuclear ketone: 3,6,8-trilydroxy-naphtalen-1-one.

The IR-spectra (Figure 8) of MSE showed the presence of bands at 3480, 3290 cm^−1^ for OH and NH, 3080, 2890 cm^−1^ for C-H aliphatic and aromatic and at 1735, 1715 cm^−1^ C=O for ester and ketone, in addition a band at 1150 cm^−1^ was characterized for the -O-ester group.

## 3. Discussion

The tested food samples were chosen from Egyptian products that are commonly used in Egypt. The results employed herein demonstrated that the staphylococci bacteria, in general, and *S. aureus,* in particular found in the quickly processed foods such as beef, chips, and corn flakes products. Since *S. aureus* strain is a pathogenic bacterium, it was of interest to concentrate in this investigation on this pathogen rather than total staphylococci bacteria. The examined food samples showed themselves to be polluted with *S. aureus* bacterium (18% of tested foods), and this was interesting and showed that there is a need to continue to research annually to make updates about the microbial pollution of foods in Egypt to give certain attention to be careful with foods [9,17,18,19,20,21,22].

The presence of MDR *S. aureus* in foods that appeared herein may be due to food preparation by hand in final packaging, and this direct contact may lead to an increase of contamination with such *S. aureus* [1]. The results of this study indicated that probably there were some poor handling hygiene during the manufacturing process of beef, chips, and corn flakes products which require more attention. The standard acceptable levels of total viable counts of *S. aureus* are 5 × 10^3^ log CFU/g [23]. The counts of presumptive *S. aureus* appeared in this study in both beef, luncheon, and potato chips are higher than this value. The problem is the possibility of resistance of the food-borne pathogen (*S. aureus*) to antibiotics. This clearly showed that there is a need to continue research to find certain natural agents to inhibit MDR *S. aureus* bacteria which exist in ready-to-eat foods. The most probable cause of high microbial count in beef processed meat might be the low hygienic quality of raw meat, insufficient storage, and thawing conditions, contamination from grinder, and the time between mincing and mixing [6,24]. 

The 18 presumptive *S. aureus* bacterial strains obtained herein gave positive results regarding catalase reaction; they were Gram-positive coccoid cells. The identification of those 18 presumptive *S. aureus* bacterial strains using manual biological tests could give elusive results [18,25]. Therefore, the identification of these bacteria was carried out by API kits (Biomerieux, Montalieu-Vercieu, France), which approved that the 18 presumptive *S. aureus* bacteria belong to *S. aureus*. These API-identification kits were used successfully for bacterial identification [8,10,11]. Identification of bacteria using API-kits is a well-established method for characterization and classification of bacteria to the species level as API-strips give accurate identification based on standardized extensive database (APIWEB^TM^ serve) in safe and quick procedures as provided by Biomereux Company (Montaliea, France) [1,8,10].

*S. aureus* No. B3 strain was showed to be methicillin resistant (MRSA), and this was confirmed at the molecular level as MRSA linkage gene, *mec A* was found to be located in *S. aureus* B3 genome [26]. Since *S. aureus* No. B3 was resistant to 10 out of 12 antibiotics tested, it was necessary to identify such strain at the molecular level by the sequencing of its 16S rRNA gene of its genome; this was necessary to get a map for this strain describing its phenotypic and genotypic characterization. Consequently, the sequence of 16S rRNA gene and its comparison with that stored in Gene Bank approved the identification given by API-kits. This strain was designated *S. aureus* LC 554891. In addition, it was found that the *S. aureus* LC 554891 strain contained both *tsst-1* and *fnb5* genes encoding for toxic shock syndrome and fibrinogen gene, respectively. This showed that the virulence factors of such LC 554891strain concur with further published work in this respect [4,27]. This LC 554891 strain was showed to contain both *hla* and *hlb* genes encoding for both α and β blood hemolysis, respectively, and this showed the interest in such B3 strain [28]. The enterotoxins genes (*sea, seb, sec*) were also found in the genome of the strain LC 554891. This shows more interest to find out natural and safe agents which could inhibit such pathogen either singly or in combinations with antibiotics. In this regard, essential oils of the plants garlic, moringa, and clove, either MSE or MLE and HB, were tested for their inhibitory activity against such MDR *S. aureus* Lc554891 (no B3) in this study.

Essential oils of plants have been used for many thousands of years in food preservation. It is necessary to investigate those plants to improve the quality of healthcare. The novel antimicrobial compounds from essential oils are potential inhibitors against bacterial pathogens [10].

Previous results have showed that *M. oleifera* seed extracts inhibited pathogenic bacteria; inhibition zone diameters were greater than 6 mm [29]. The aqueous extract of both MSE and MLE were found to be strong inhibitory against the standard *S. aureus* ATCC6538 and *S. aureus* LC 554891; the diameter of inhibition zone appeared to increase with increasing the concentration of the antibacterial agents used. In the results employed herein, aqueous extracts of both MSE and MLE showed the better inhibitory activity against *S. aureus* LC 554891. The reason behind this might be due to the fact that the aqueous extraction of the bioactive substance did not alter the structure of such compounds and keep them active [10].

In view of the bioactive compounds elucidated by GC-MS spectroscopy, almost all of them were reported to inhibit bacterial pathogens by different mechanisms of action [18]. Esters and Ketones are, in general, positively charged and more hydrophobic; such hydrophobocity allows electrostatic interactions with the bacterial cellular components, leading to a loss of cell viability due to the formation of fully de-energized killed cells [9]. The 6-iodoacetoveratrone elucidated in this study appeared also in a previous study to inhibit different indicator bacteria by almost similar mechanisms [30,31]. The octadeca-anoic acid that appeared herein showed itself to be antibacterial because acids decrease pH value to levels where *S. aureus* cannot grow [31,32]. Previous studies have showed that bisethylhexyl phthalate and polynuclear ketones inhibited methicillin resistant *S. aureus* by an almost similar mode of action [4,5,33]. Heterocyclic compounds appeared herein by GC-MS analysis inhibit pathogenic bacteria cells as they can interact either electrophils or nucleophiles of the cells, leading to inhibition of DNA synthesis which causes cell death [34]. Finally, aromatic amines were reported to interact with DNA directly through the formation of covalent adducts [30,33]. It will be necessary to test the antimicrobial activity of each compound alone. 

This also might be due to osmotic pressure of the solutes which existed in hypertonic medium in relation to the outer aquatic medium; this facilitates the diffusion of the bioactive materials from cell membranes across the selective permeability. The lipophilic nature of some solutes facilitates their attachment to bacterial cell membranes which in turn causes cell death [8,9,35].

This study was undertaken also to investigate the in vitro antibacterial activity of honey against the selected LC 554891 strain. In this study, the honey sample showed an antibacterial activity against the *S. aureus* LC 554891 strain, and this is in agreement with previous published results [36]. Such results are in confirmation with Mama et al. [37], who declared that the inhibitory activity of honey bees is due to a mix of antibacterial agents such as high content of hydrogen peroxide, powerful antioxidants, naturally low pH, which is unsuitable for bacterial growth, and to the presence of phenolic acids, lysozymes, and flavanoids. The potency of native honey (100% concentration) was found to be inhibitory against *S. aureus* LC 554891, and this concentration was the best one giving antibacterial activity; such results concur with previously published work [38]. A previous study [39] discovered that the antimicrobial activity of honey was more with *S. aureus* and *Acinetobacter* spp, both with resistance to some antibiotics like gentamicin, ceftriazone, amikacin, and tobramicin than other bacteria tested. Honey bees have an antibacterial nature due to presence of H2O2, phenolic compounds, and pH [10]. There are polyphenolic compounds present in honey bees, which are responsible for its antibacterial activity. The common polyphenolic compounds are gallic acid, cinnamic acid, ferulic acid, hydroxyl cinnamic acid, sinapic acid, syringic acid, and chlorogenic acid. These compounds inhibit the bacteria by disrupt bacterial membrane, inhibit DNA gyrase, induce topaisomerase IV mediated DNA cleavage, inhibit peptidoglycan, and ribosome synthesis [40].

Due to the promising inhibition of *S. aureus* LC 554891 strain by MSE, a mixture of this MSE and the antibiotic tetracycline was used as an inhibitory agent for this B3 strain. The vigorous and distractive inhibition of *S. aureus* LC 554891 strain showed an interesting perspective to use such mixture as a biocontrol agent for *S. aureus* [10,11,40].

The combinations of both MSE and tetracycline gave broader antibacterial activity because both of them may be acted in a synergism; a synergism between antibiotics and plant extracts could be due to binding of both of them by hydrogen bonding, hydrophobic-hydrophobic interactions, and molecular interactions (10). In view of the tetracycline molecule, it contains four fused rings (A, B, C&D) to which a variety of polar groups (5- hydroxyl groups and one amino group) are attached. In addition, the bioactive compounds of MSE elucidated herein contain both polar and non-polar chemical moieties. Hence, an interaction of MSE and tetracycline might occur between polar and non-polar chemical moieties [41]. In fact, further experiments will be needed to study such synergism at the molecular chemical level. 

Many MDR bacteria including methicillin resistant *S. aureus* (MRSA) can be controlled using pharmaceutical potentials in the treatment of infection for example, Bis [2-ethyl hexyl] phlthalate [42]. 1-(Hydroxymethyl)-1,2-etheraneelyl ester octadecanoic acid is known to have antibacterial activity and is thought to play a more direct role than previously thought in innate immune defense against epidermal and mucosal bacterial infections [43,44]. The anti-staphylococcal activity of MSE is due to the mix of the compounds that appeared herein from GC-MS. Similarly, a previous study that bioactive compounds of *M*. *oleifera* seed extract e.g., 4-(α-l-rhamnopyranosyloxy) benzyle isothiocyanate strongly inhibited the *S. aureus* BAA-977 strain [45]. This means that *M*. *oleifera* could be used as a safe and potent control of infectious diseases [45]. Extracts of *M*. *oleifera* leaf, stem, and seeds were used as inhibitory agents against *S. aureus* bacteria isolated from human sputum [46].

With regard to the mechanism of action of the bioactive compounds that appeared from the instrumental analysis used in this study in MSE, the antibacterial activity of fatty acids are attributed to their ability to disrupt the outer bacterial cell membrane, increasing the leakage of electrolytes from bacterial cells and, in turn, cause cell death [47]. In addition, it was approved that the other bioactive compounds of *M*. *oleifera* seed extracts such as minerals, aromatic amines, esters, and ketones inhibit cell wall synthesis of bacteria at its initial stages and accumulate onto a cell membrane, leading to interruption in the bacterial metabolism and cell death [48,49].

It is necessary to test the anti-staphylococcal activity of each compound alone. Studies in this regard are under investigations. Further work will be necessary to isolate the bioactive compounds obtained from MSE and to check the antimicrobial potential for each of them either separately or in combination with antibiotics.

## 4. Materials and Methods

### 4.1. Food Sampling

The foods used were ready-to-eat beef luncheon (Egyptian made), potato chips, and corn flakes. A hundred food samples including beef luncheon (*n* = 25), potato chips (*n* = 50), and corn flakes (*n* = 25) were examined in this study. These food samples were purchased from different retail supermarkets of urban areas of the Egyptian cities viz. Belbeis, Zagazig, Abo-Kabir, and Hehia; all of these cities are located in the Sharkia Governorate (80 km north, Cairo, Egypt). All of these food types were made in Egyptian companies. The samples were transported immediately in sterile plastic bags (Gomhuria Company, Zagazig, Egypt) to the laboratory of Microbiology Department, Faculty of Science, Zagazig University, City, Egypt. The samples (25 g) were taken under aseptic conditions to a blender (Gomhuria Company, Zagazig, Egypt), dissolved in 225 mL of sterile buffer peptone water 1–10 dilutions (0.1% *w*/*v*) and mixed well for 60 s at 25 °C.

### 4.2. Isolation of Bacteria Suspected to Be S. aureus

Serial two-fold dilutions of up to 10^−6^ were made from the initial dilution (1:10) and 0.1 mL aliquots of these dilutions were inoculated onto Baird Parker agar (Oxoid). The bacteria suspected to be *S. aureus* count were found by examining the plates at typical black colonies, convex shape, with a shiny halo zone, and these were checked for positive Gram coagulase reaction and catalase test (Bactident Coagulase Biolife, Milan, Italy). The identification of the bacterial isolates was then carried out by API-Kits (Biomereux, Montaliea, France) as given by the manufacturer’s instruction.

### 4.3. Antibiotics Susceptibility Test

The susceptibility of *S. aureus* ATCC 6538 strain (control), *S. aureus* (*n* = 18) strains to 12 antibiotics were tested by standard disc diffusion technique CLSI [47]. The cultures were grown in nutrient broth (Oxoid) for 12 h. Inocula were adjusted at 10^5^ CFU/mL and then were plated (100 µL per plate) onto Muller Hinton agar (Hi-Media, Mumbai, India). The following antibiotic discs with their concentrations indicated in parenthesis were used (All from Johnson & Johnson, Egypt. Branch, Heliopolis, Cairo, Egypt) viz. spiramycin (SP: 100 µg), clindamycin (DA: 2 µg), doxycycline (DO: 30 µg), ampicillin (AM: 10 µg), tetracycline (TE: 30 µg), ciprofloxacin (CIP: 5 µg), neomycin (N: 30 µg), ofloxacin (OFX: 5 µg), amoxicillin (AMC: 30µg), penicillin G (P: 10 µg) oxacillin (OX: 10µg), methicillin (ME: 5 µg). The antibiotic discs were placed onto Muller Hinton agar plates that seeded with the tested bacteria; plates were then inverted and incubated at 37 °C for 24 h. Results were expressed by measuring inhibition zone diameters (IZDs) by millimeters. Multiple antibiotic resistance index was calculated by using the following formula: MAR Index = Number of antibiotics to which the isolate was resistant/Total number of antibiotics tested. Spiramycin (SP: 100 µg), Clindamycin (DA: 2 µg), Doxycycline (DO: 30 µg), Ampicillin (AM: 10 µg), Tetracycline (TE: 30 µg), Ciprofloxacin (CIP: 5 µg), Neomycin (N: 30 µg), Ofloxacin (OFX: 5 µg), Amoxicillin (AMC: 30 µg), Penicillin G (P: 10 µg) Oxacillin (OX: 10 µg), Methicillin (ME: 5 µg) [50,51].

### 4.4. Molecular Identification of S. aureus No. B3

It was necessary to confirm the identification of the MDR *S. aureus* No. B3 bacterium by fingerprinting of the sequence of 16S rRNA gene. Hence, DNA was extracted from the B3 strain [52]. The 16S rRNA gene was amplified by PCR with using universal primers (forward primer [F27] 5’-AGAGTTTGATCCTGGCTCAG-3’ [53] and reverse primer [R1492] 5’-GGTTACCTTGTTACGACTT-3’) [54]. The PCR was carried out in a Gene-Amp PCR system 9600 thermocycler (Perkin Elmer Co., Jersey, AL, USA). The amplification conditions were as follows: 94 °C for 10 min and 35 cycles of denaturation at 95 °C for 30 s, annealing-extension at 56 °C for 1 min, 72 °C for 1 min and an extension at 72 °C for 10 min. The PCR product was electrophoresed using agarose gel (0.7%) (Gomhuria, Egypt). The 16SrRNA gene band appeared at 1500 bp was cut by Gene Purification Kit (Promega Corporation, Madison, WI, USA). 

The nucleotide sequence of 16S rRNA gene of the *S. aureus* LC 554891genome was sequenced by using 3130 X DNA Sequencer (Genetic Analyzer, Applied Biosystems, Hitachi, Ibaraki, Japan) as described previously [55,56]. The way to record such strain in gene bank included the submission of the sequence of 16 S rRNA gene of the B3 genome to Gene Bank at http://blast.ncbi.nlm.nih.gov/Blast.cgi?PROGRAM=blastn&PAGE_TYPE=BlastSearch&LINK_LOC=blasthome. By using the Basic Local Alignment Search Tool Program, a phylogenic tree and cluster analysis were carried out by clusta 1× Program for estimation of the similarity between the isolated strains and the stored *S. aureus* strains in the database. It was shown clearly that the B3 strain is similar by ˃ 99.5% to *S. aureus* category (Figure 2). An accession Number LC554891 on the NCBI web server (http://blast.ncbi.nlm.nih.gov/Blast.cgi?PROGRAM=blastn&PAGE_TYPE=BlastSearch&LINK_LOC=blasthome) of such strain was given as a record code for this strain. Consequently, such strain was designated as *S. aureus* LC554891. By using the Basic Local Alignment Search Tool Program, a phylogenic tree and cluster analysis were carried out by clusta 1× Program for estimation of the similarity between the isolated strains and the stored *S. aureus* strains in the database.

### 4.5. Detection of Virulence Factors (sea, seb, sec, tsst-1 and fnbA) of the Strain

Extraction of DNA from *S. aureus* LC 554891 was performed using DNeasy bacteria Mini Kit (Bio Basic Comp., Toronto, ON, Canada) [57,58]. PCR was performed in 30 µL volume tubes according to Williams et al. [57]. The DNA amplifications were performed in an automated thermal cycle (Promega Corporation, Madison, WI, USA) programmed for one cycle at 94 °C for 4 min followed by 10 cycles of (4 min at 94 °C, 1 min at 52A °C, and 1 min at 72 °C) then 15 cycles of (4 min at 94 °C, 1 min at 58 °C, and 1 min at 72 °C) the reaction was finally stored at 72 °C for 10 min. The DNA amplified product (15 µL) was loaded in each well of agarose gel electrophoresis equipment (Gomhuria, Egypt) using DNA ladder (100 bp) mix that used as standard DNA with known molecular weights. The run was performed for about 30 min at 80 V in mini submarine gel (Bio-Rad Laboratories, Berkeley, CA, USA). PCR reactions were conducted using 4 simple Sequence Repeat (SSR) primers. Their names and sequences are shown in Table 5.

### 4.6. Genetic Linkage of mecA, hla, and hlb Genes

Total DNA was extracted from exponentially growing B3strain cells [52,59]. The *mecA* gene primers were *mecA* f (AAAATCGATGGTAAAGGTTGGC) and *mecA r* (AGTTCTGCAGTACCGGATTTGC) [60]. In addition, primers used for *hla* gene were *hla f* GCC AAA GCC GAA TCT AAG and *hla r* GCG ATA TAC ATC CCA TGG C [61] and those used for *hlb* gene were *hlb f* TTGGCTGGGGAGTTGAAGCACA and hlb r CGCCTGCCCAGTAGAAGCCATT (Promega Corporation, Madison, WI, USA) [62]. PCR rounds were carried out by using 5µl of template DNA, 0.025µM of each primer, (Promega Corporation, Madison, WI, USA). DNA amplification was carried out for 40 cycles in 100 µl of reaction mixture as follows: denaturation of 94 °C for 30 s, annealing at 55 °C for 30 s, and extension at 72 °C for 1 min with a final at 72 °C for 5 min. Ten micro liters aliquots of PCR products were analyzed using 1.5% agarose gel electrophoresis at 90 V (Gomhuria Company, Zagazig, Egypt) for 90 min.

### 4.7. Screening of the Antibacterial Activity of Essential Oils of Moringa olifera, Allium sativum, and Syzygium aromaticum against S. aureus LC 554891

Essential oils of *Moringa olifera, Allium sativum and Syzygium aromaticum* were obtained from El-Hawag factory, Bader, Egypt, under the supervision of Ministry of Health license no: 150/80 for the year 2002. Then, they sterilized with 0.45 μm filter paper obtained from a High Lab Company, Zagazig, Sharkia, Egypt. Sterilized filter paper discs (6 mm diameter) were soaked in 1 mL of each essential oil used, for 2 min. They were then placed onto BHI agar plates that were inoculated by cell suspension of *S. aureus* LC 554891. After incubation for 24 h at 37 °C, diameter of inhibition zones (mm) were measured after subtracting diameter of paper disc [50].

### 4.8. Preparation of the M. oleifera Leaves (MLE) and Seeds (MSE)

The plant *M. oleifera* was identified by the plant taxonomist, Prof. Dr. Hussein Abdel-Basset, at Department of Botany and Microbiology, Faculty of Science, Zagazig University, Egypt. Both *M. oleifera* leaves and seeds were collected; the leaves were cleaned from extraneous matter and properly washed then dried in hot air oven (Alexandria Co., Alexandria, Egypt) for 24 h at 40 °C. The seeds were dried and grounded to powdered form using a clean sterile mortar and pestle (Moulinex, Cairo, Egypt) and packaged in an air tight plastic container (Alexandria Co., Alexandria, Egypt) until used. About 10 g aliquots of either powdered leaves or seed-powdered were macerated in 100 mL distilled water and allowed to be extracted for 48 h at room temperature; methanolic and ethanolic extracts were also carried out by homogenization of either MSE or MLE (10 g for each) with 100 mL ethanol or methanol for 40 min [11]; solvents were then evaporated by keeping the extracts in an oven (Alexandria Co., Alexandria, Egypt) adjusted at 60 °C overnight. Both leave extracts (MLE) and seed extracts (MSE) were homogenized with sterile water and sterilized by filtration (0.45 milipore Bilters, Amicon, Mumbai, India). Stock preparation of MSE (200 μg/mL) was prepared and then stored in Eppendorf tubes (Gomhuria Co., Zagazig, Egypt) at 5 °C until antimicrobial activity tests were performed [63].

### 4.9. Preparation of Honey Bee (HB) Solutions

Native HB used in this study was provided by a bee-keeper from Kafr-Sakr area, Sharkia Governorate (104 km North Cairo), Egypt. It was aseptically collected in 100 mL screw capped bottles, transported to the laboratory. HB dilutions were prepared immediately prior their testing by diluting native honey to the required concentrations (10%, 20%, 30%, 40%, 50%, 60%, 70%, 80%, 90% *v*/*v*) [10,21]. These dilutions were made using sterile distilled water. A series of measurable scaled 250-mL screw capped bottles (Gomhuria Company, Zagazig, Egypt) containing 90 mL; 80 mL; 70 mL; 60 mL; 50 mL; 40 mL; 30 mL; 20 mL; and 10 mL native HB were prepared and completed to 100 mL sterile distilled water using sterile pipettes, giving the desired dilutions of HB viz. 10%, 20%, 30%, 40%, 50%, 60%, 70%, 80%, and 90%, respectively.

### 4.10. Bioassay of the Antibacterial Activity of MLE, MSE, and HB

Brain Heart infusion agar plates (DifcoTM, Maryland, MD, USA) were prepared and seeded log phase cells (10 **^5^** CFU/mL); then, the sterile natural agents concentrations listed in Table 3 were added by automatic pipette to these filter paper discs (6 mm diameter), which were placed immediately onto the above plates. The controls were filter paper discs soaked in sterile distilled water. Samples and controls were incubated at 37 °C for 24–48 h. IZDs were measured after subtracting the diameter of the paper disc [17,21,64].

### 4.11. Minimum Inhibitory Concentration (MIC) of the MSE Extract

A stock prepared MSE contained 200 µg/mL. From this MSE concentration, different dilutions were made to contain 10, 20, 30, 40, 50, 60, 70, 80, and 90, µg/mL, respectively. Serial two-fold dilutions of MSE were made in sterile deionized water by taking 0.05, 0.1, 0.15, 0.2, 0.25, 0.3, 0.35, 0.4, and 0.45 mL, from the original stock and this equals 10, 20, 30, 40, 50, 60, 70, 80, 90 µg/mL, respectively. Then, sterile filter paper discs were saturated with the MSE dilutions and placed onto Muller Hinton agar (DifcoTM, Maryland, MD, USA) that seeded previously with activity growing cells of *S. aureus* B3. The antibacterial activity was studied by a disc diffusion assay as described above. MIC was visually identified as the lowest concentration of MSE that inhibited bacterial growth [9,10,11,40].

### 4.12. Antibacterial Activity of Combination of Antibiotics and MSE

The antibiotic tetracycline listed in Table 2 that inhibited the *S. aureus* strain was mixed with MIC value of MSE. Sterile filter paper discs were impregnated by these combinations and assayed for their antistaphylococcal activity as described above. In addition, single different concentrations of either tetracycline or MSE were tested singly for their antistaphylococcal activity. Mixtures of MSE with the antibiotic tetracycline were made as follows: (20 µg/mL MSE + 10 µg tetracycline) and (40 µg/mL MSE + 10 µg tetracycline). Filter paper discs of 6 mm diameter were soaked in each combination and the experiment was carried out as described above [8].

### 4.13. Instrumental Analysis of MSE

To determine and identify the bioactive compounds of MSE, Gas Chromatography–Mass Spectroscopic (GC-MS) was used (Trace GC 1310-ISQ Mass Spectrometer, Thermo Scientific, Austin, TX, USA). A direct capillary column TG–5MS (30 m × 0.25 mm × 0.25 µm film thickness) was used. About 3 µL of MSE was injected automatically to the equipment using Auto sampler AS3000 coupled with GC in the split less mode. Then, the instrumental analysis was carried out as described previously [15,65]. The components were identified by comparison of their retention times and mass spectra with those of WILEY 09 and NIST 11 mass spectral database [66]. 

### 4.14. Statistical Analysis

All the experiments were performed in triplicates and results were expressed by the mean with the standard error. Data were statically analyzed using ANOVA variance analysis (SAS version 9.1, SAS Institute, Inc., Cary, NC, USA) [67]. Basic Local Alignment Search Tool Program (BLAST) was used to construct the pairwise similarity of the *S. aureus* B3 with *S. aureus* cluster of Gene Bank; Clusta 1X Tree Programme was used to construct the phylogenetic tree.

### 4.15. Ethical Approval

This work was approved by institutional review board at Faculty of Science, Zagazig University, Zagazig, Egypt.

## 5. Conclusions

Some ready-to-eat Egyptian food showed itself to be polluted with *S. aureus* bacteria. The obtained bacteria were studied regarding their susceptibility to different antibiotics; one strain appeared to resist the action of 10 antibiotics of 12 ones tested; this strain was characterized at the molecular level for its virulence capability. Different available natural agents were tested for their inhibitory action against *S. aureus* LC 554891. MSE inhibited distinctively such strain. A combination of MSE and the antibiotic tetracyclin appeared to be a powerful inhibitory agent against *S. aureus* LC 554891.

## Figures and Tables

**Figure 1 molecules-25-04583-f001:**
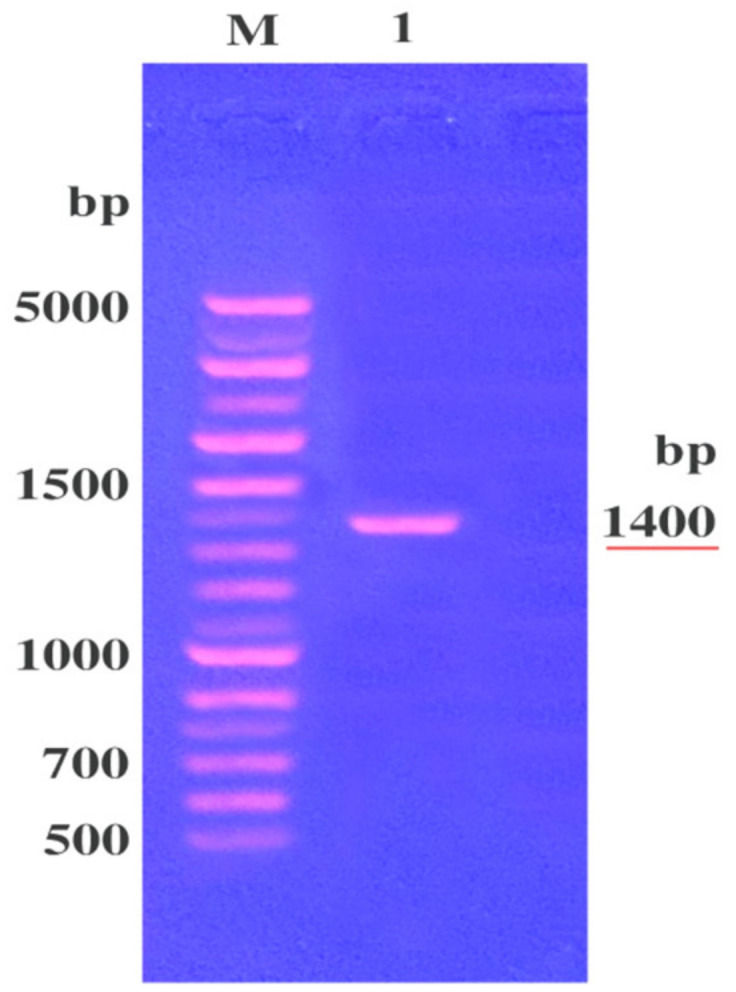
Agarose gel electrophoresis of amplified DNA (PCR product) obtained from 16S rRNA gene of *S. aureus* B3 (MRSA B3). Lane M, DNA marker of known molecular sizes; Lane **1**, PCR product of the amplified 16S r RNA gene of *S. aureus* B3.

**Figure 2 molecules-25-04583-f002:**
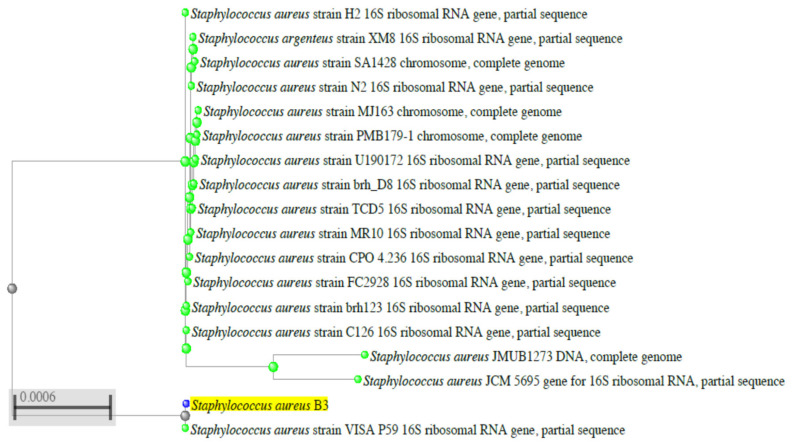
Cluster and dendrogram analysis showing the phylogenetic tree of *S. aureus* B3 with 99.5% similarity with a *S. aureus* cluster.

**Figure 3 molecules-25-04583-f003:**
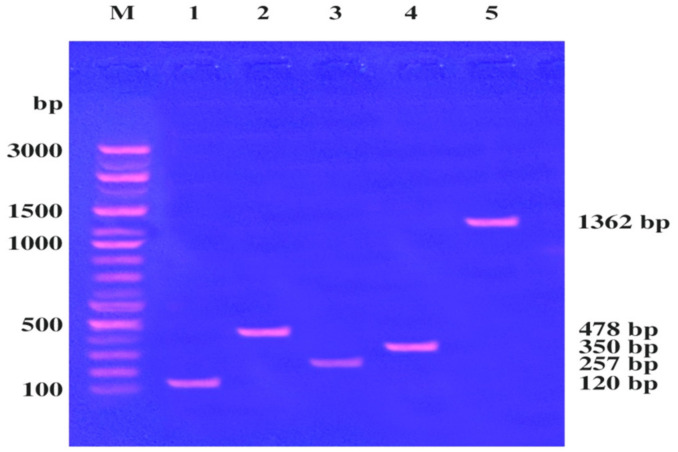
Detection of *S. aureus* B3 virulence factors in two multiplex PCR reactions; DNA marker (M), *sea* (1), *seb* (2), *sec* (3), *tsst-1* (4), and *fnbA* (5).

**Figure 4 molecules-25-04583-f004:**
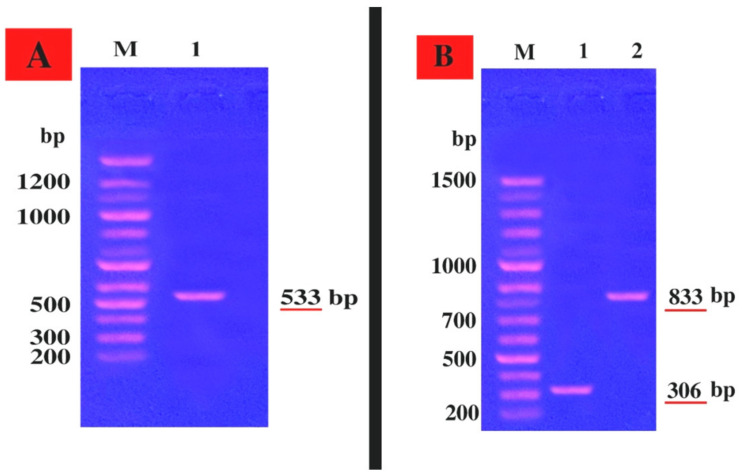
Agarose gel electrophoresis of both *mecA*, *hla*, and *hlb* genes. (**A**) *mecA* gene, Lane **M**, marker DNA of known molecular sizes; Lane (**1**) PCR product of *mec A* gene. (**B**) *hla* and *hlb* genes; Lane **M**, marker DNA of known molecular sizes; Lane (1) PCR products of *hla* gene, Lane (2) PCR products of the *hlb* gene.

**Figure 5 molecules-25-04583-f005:**
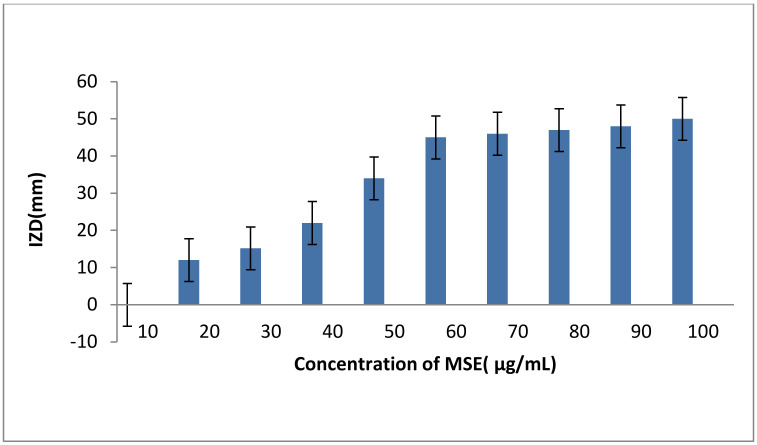
Determination of Minimum inhibitory concentration (MIC) of MSE against *S. aureus* LC 554891. All values reflect the mean values of three replicates and standard deviations.

**Figure 6 molecules-25-04583-f006:**
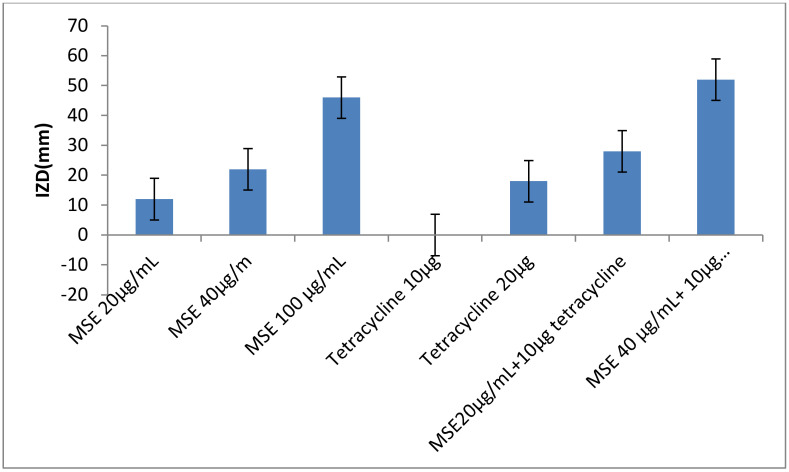
Inhibition of *S. aureus* LC 554891 by combinations of both tetracycline and MSE. All values reflect the mean values of three replicates and standard deviations.

**Figure 7 molecules-25-04583-f007:**
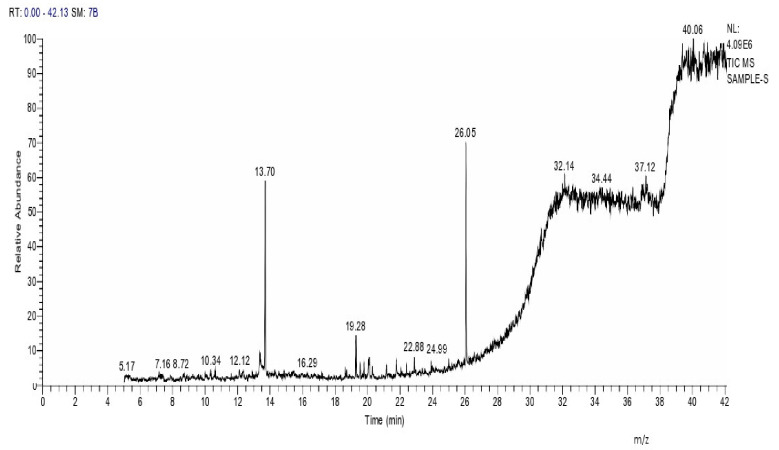
GC-MS analysis of MSE.

**Figure 8 molecules-25-04583-f008:**
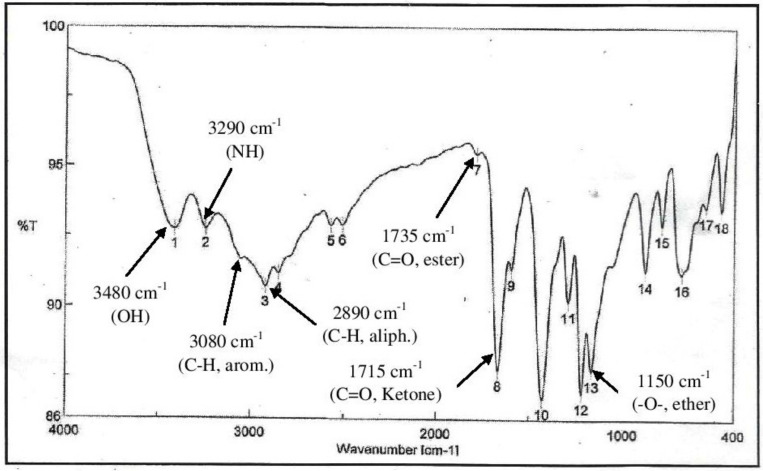
IR spectrum in KBr (discs) for the extraction of MSE.

**Table 1 molecules-25-04583-t001:** Antibiotic sensitivity test of the studied bacteria based on the diameter of inhibition zone (mm).

Strain	Diameters of the Inhibition Zone (mm) ± SD												MAR Index
	SP	DA	DO	AM	TE	CIP	N	OFX	AMC	P	OX	ME	
*S. aureus* ATCC 6538	0	0	0	0.8 ± 0.10	0	2.1 ± 0.15	1± 0.07	0	0.8 ± 0.11	1.7 ± 0.18	1 ± 0.09	0.9 ± 0.13	5 (41.67%)
*S. aureus B3*	0	0	0	0	4± 0.36	3 ± 0.25	0	0	0	0	0	0	10 (83.3%)
*S. aureus B7*	1	0.9 ± 0.08	2.9 ± 0.18	0	3 ± 0.29	2.1 ± 0.16	0	0	1.2 ± 0.08	1.7 ± 0.15	0	0.9 ± 0.10	4 (33.33%)
*S. aureus B8*	0	2.9 ± 0.26	1.5 ± 0.15	0	0	3 ± 0.34	0.8 ± 0.10	0	1 ± 0.10	0	0	0	7 (58.33%)
*S. aureus B14*	0	0.9 ± 0.10	2.9 ± 0.23	0	1	2.1 ± 0.20	o.8 ± 0.11	0	1.6 ± 0.21	0	0	0.9± 0.11	5(41.67%)
*S. aureus B17*	1.1 ± 0.09	0.8 ± 0.12	0	0	0	2.1 ± 0.22	1.1 ± 0.14	0	0	0	2.9 ± 0.32	1.7 ± 0.15	6 (50%)
*S. aureus B18*	2.1 ± 0.20	1.1 ± 0.13	0	2.9 ± 0.32	0	0	0	0	1.1 ± 0.15	1.7 ± 0.16	2.1 ± 0.39	1.1	5 (41.67%)
*S. aureus B22*	2.1 ± 0.22	0.8 ± 0.10	2.1 ± 0.20	0	0	2.1 ± 0.25	0.9 ± 0.09	0	1 ± 0.12	0	2.9 ± 0.40	1.7 ± 0.15	4 (33.33%)
*S. aureus B24*	2.9 ± 0.30	1.7 ± 0.14	0	0	0	2.1 ± 0.23	0.8 ± 0.10	0	1.2 ± 0.13	0	0	0	7 (58.33%)
*S. aureusCh32*	2.9 ± 0.31	0	2.9 ± 0.33	2.9 ± 0.35	0	0	0	0	1 ± 0.09	1.7 ± 0.18	2.1 ± 0.28	1 ± 0.10	5 (41.67%)
*S. aureus Ch35*	1.1 ± 0.12	2.1 ± 0.26	0	2.9 ± 0.32	0	2.1 ± 0.22	1 ± 0.13	0	0	1.7 ± 0.16	1 ± 0.10	1.1 ± 0.13	4 (33.33%)
*S. aureus Ch40*	0.8 ± 0.10	0	0.8 ± 0.09	2.9 ± 0.39	0	0	0	0	1 ± 0.10	2 ± 0.22	0	0.9 ± 0.09	6 (50%)
*S. aureus Ch41*	0	3.2 ± 0.27	0.9 ± 0.10	3.2 ± 0.25	0	0	0	0	0	2 ± 0.23	0	0.9 ± 0.09	7 (58.33%)
*S. aureus Ch48*	1.2 ± 0.07	1 ± 0.1	0	2.9 ± 0.31	0	2.1± 0.22	1 ± 0.11	0	0	1.7 ± 0.15	0.9 ± 0.07	1 ± 0.10	4 (33.33%)
*S. aureus Ch50*	1.2 ± 0.07	0	2.1 ± 0.19	2.9 ± 0.33	0	0	0	0	1 ± 0.11	1.7 ± 0.16	0	0	7 (58.33%)
*S. aureus Ch53*	2.7 ± 0.30	0	0	0	0	0	0	0	1.2 ± 0.10	1.7 ± 0.15	2.7 ± 0.25	1 ± 0.10	7 (58.33%)
*S. aureus Cf58*	1.2 ± 0.11	0	0	3.2 ± 0.33	0	0	0.9 ± 0.08	0	0	2 ± 0.25	3.4 ± 0.41	1.3 ± 0.08	6 (50%)
*S. aureus Cf66*	3.5 ± 0.25	0.9 ± 0.88	0	2 ± 0.19	0	3.5 ± 0.30	0.9 ± 0.10	0	0	1.3 ± 0.11	0	0.9 ± 0.08	5 (41.67%)
*S. aureus Cf69*	1 ± 0.09	3.2 ± 0.29	2 ± 0.18	0	0	3.5 ± 0.33	0.9 ± 0.12	0	1.3 ± 0.11	0	0	0.9± 0.08	5 (41.67%)
No. of strains (%)	5(19) 26.3%	7(19) 36.8%	10(19) 52.6%	9(19) 47.4%	16(19) 84.2%	7(19) 36.8%	8(19) 42.1%	19(19) 100%	8(19) 42.1%	7(19) 36.8%	10(19) 52.6%	4(19) 21.1%	

For strain designation B; Ch; Cf, refer to the fact that these are strains isolated from beef luncheon; potato chips; corn flakes, respectively.

**Table 2 molecules-25-04583-t002:** The effect of some essential oils such as garlic, moringa, and clove oils against the selective *S. aureus* strain LC 554891 by using a disc diffusion method compared to tetracycline (10 µg/disc).

Diameters of Inhibition Zone (mm) ± SD							
Strains	Tetracycline 10 µg/disc (Positive Control)	Garlic Oil		Moringa Oil		Clove Oil	
		0.25%	0.5%	0.25%	0.5%	0.25%	0.5%
*S. aureus* ATCC 6538	16.0 ± 0.0	23 ± 0.2	31 ± 0.2	35 ± 0.1	37 ± 0.1	14 ± 0.1	23 ± 0.0
*S. aureus* LC 554891	0	25 ± 0.1	32 ± 0.2	30 ± 0.1	35 ± 0.2	13 ± 0.3	24 ± 0.1

**Table 3 molecules-25-04583-t003:** Inhibitory activity of both *Moringa oleifera* extracts (either leaves (MLE) or seeds (MSE) and honey bee (HB) against *S. aureus* LC 554891.

The Bacteria	Diameters of Inhibition Zone (mm) ± SD																					
	Tetracycline (10 µg) (Positive Control)	MLE (µg/mL)									MSE (µg/mL)									HB (%)		
		Methanolic Extract (ME)			Ethanolic Extract (EE)			Water Extract (WE)			Methanolic Extract (ME)			Ethanolic Extract (EE)			Water Extract (WE)					
		50	100	200	50	100	200	50	100	200	50	100	200	50	100	200	50	100	200	5	50	100
*S. aureus* ATCC 6538	15.0 ± 0.0	9.5 ± 0.31	11 ± 0.00	13.0 ± 0.91	12.0 ± 0.7	13.1 ± 0.47	14.7 ± 0.7	5.6 ± 0.24	8.3 ± 0.91	9.3 ± 0.46	12.00 ± 0.00	15.2 ± 00.1	18.81 ± 0.61	8.1 ± 0.00	9.31 ± 0.07	10.00 ± 0.00	34 ± 0.11	47 ± 0.00	50.00 ± 0.00	0	30.1 ± 0.2	34.43 ± 0.00
*S. aureus* LC 554891	0	8.3 ± 0.7	9.3 ± 0.46	11.0 ± 0.00	10.0 ± 0.7	10.0 ± 0.7	13.0 ± 0.00	2.3 ± 0.00	5.6 ± 0.00	6.4 ± 0.7	11.00 ± 0.98	14.25 ± 0.12	17.75 ± 0.41	8.0 ± 0.00	9.0 ± 0.07	10.00 ± 0.00	32 ± 0.11	45 ± 0.11	48.00 ± 0.00	0	30.51 ± 0.07	34.00 ± 0.00

All values reflect the mean values of three replicates and standard deviations (-): No inhibition zone, ME: methanolic extract; EE: ethanolic extract; WE: water extract. MSE: *M. oleifera* seed extract; MLE: *M. oleifera* leaves extract; HB: Honey bee.

**Table 4 molecules-25-04583-t004:** Putative identification of 14 components from MSE when subjected to GC-MS (gas liquid chromoatographic mass spectrometry).

No.	Classification	M. Formula	M.W.	Compound Name and Structure	Area	Parent Ion (M^+^)	Base Peak (m/e) (100%)
1	Spiro ketone	C_17_H_24_ O_3_	276.0	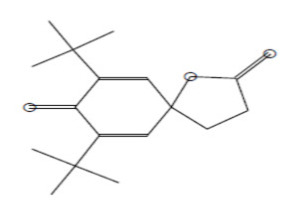 7,9-Di-tert-butyl-1-oxaspiro [4,5] deca-6,9-diene-2,8-diene	5.25	276.0	57.00
2	Ester	C_18_H_28_O_3_	292.0	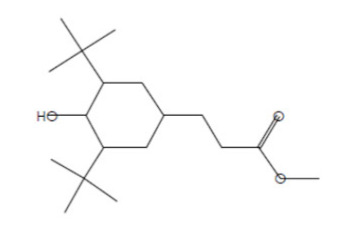 Methyl 3-[3,5-di-tert-butyl-4-hdroxy phenyl] propionate	2.04	292.0	277.0
3	Heterocyclic compounds	C_14_H_22_N_2_O	234.0	1-Methyl-2-cyano-3-ethyl-4-pivaloyl-2-piperidine	2.08	234.0	149.0
4	Polynuclear ketone	C_10_H_10_O_4_	194.0	3,6,8-Trilydroxy-Naphtalen-1-one	2.08	195.0(M^+1^)	149.0
5	Saturated fatty ester	C_18_H_36_O_2_	284.0	Ethyl hexadecanoate	7.12	284.0	88.0
6	Ketone	C_10_H_11_IO _3_	306.0	6-Iodoacetoveratrone	1.77	308.0(M^+2^)	291.0
7	Ketone	C_13_H_18_ O_2_	206	2-Alyl-5-t-butyl hydroquinone	27.82	207.0 (M^+1^)	191
8	Fatty Ether	C_14_H_28_O	212.0	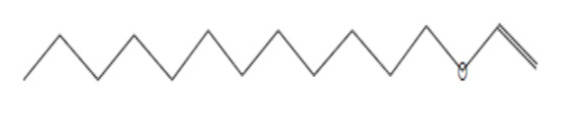 Vinyl lauryl ether	2.04	212.0	43.0
9	Fatty ester	C_16_H_32_O_3_	272.0	Hydroxy ethyl myristate	2.66	272.0	104.0&43.0
10	Fatty ester	C_18_H_36_O_3_	300.0	2-Hydroxy ethyl palmitate	2.66	300.0	104.0&43.00
11	Dibasic fatty acid	C_18_H_34_O_4_	314.0	Octadecanedoic	2.66	314.0	98.0
12	Ester	C_24_H_38_O_4_	390.0	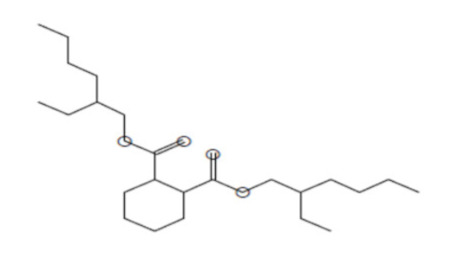 Bis [2-ethyl hexyl] phlthalate	29.30	391.0(M^+1^)	149.0
13	Fatty ester	C_39_H_76_O_5_	624.0	1-(Hydroxymethyl)-1,2-etheraneelyl ester octadecanoic acid		625.0(M^+1^)	267.0
14	Aromatic amines	C_13_H_17_NO	203.0	Formylcyclohexyl Aniline	3.43	203.0	174.0

**Table 5 molecules-25-04583-t005:** Virulence genes and the primers used for the detection of *S. aureus* LC554891 genome.

Detected Virulence Factors	Primer Sequence (Forwarded)	Primer Sequence (Reverse)	Size of the PCR Products (bp)
*Sea*	TTGGAAACGGTTAAAACGAA	GAACCTTCCGATCAAAAACA	120
*Seb*	TCGCATCAAACTGACAAACG	GCAGGTACTCTATAAGTGCC	478
*Sec*	GACATAAAAGCTAGGAATTT	AAATCGGATTAACATTATCC	257
*Tsst-1*	ATGGCAGCATCAGCTTGATA	TTTCCAATAACCACCCGTTT	350
*fnbA*	CACAACCAGCAAATATAG	CTG TGTGGTAATCAATGT	1362

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
