# Peer review of "Inhibition of Staphylococcus aureus LC 554891 by Moringa oleifera Seed Extract either Singly or in Combination with Antibiotics"

_molecules, 2020, doi:10.3390/molecules25194583_

Round 1

Reviewer 1 Report

All the suggestions are in the attached document. 

Author Response

Reviewer # 1   Round 2                                                    

I strongly recommended reviewing the writing of the manuscript.

We thank the reviewer for that point. Ok, the  manuscript is revised again regarding English and is will improved. The manuscript improved in its overall quality. The corrected parts are marked by yellow color.

Line 16: The experassion: plant extracts were used to control human pathogens is too broad…….

We thank the reviewer for  this nice comment. The sentence was written again crucially regarding the study target and is indicated in the manuscript by yellow colour.

Line 24: Please check English grammar.

Ok, it was checked, revised and marked by yellow colour.

Lines 27:  the whole paragraph has English grammar, Please check

Ok, it was checked, revised and marked by yellow colour.

Line 74: the elements should be separated by a comma and not by semicolon.

Ok, it was checked again and revised.

Line 75: A proper reference should be added.

A proper and round reference was added. (Abd Rani et al., 2018, Frontiers Pharmacology.  

Line 100-105: Review English grammer.

Ok, the whole paragraph was written again and highlighted in the text.

Table 1 should include the slandered deviation of all the inhibition halos.

Thanks for this comment. We reviewed the statistics and SD are given.

Line 116: Equation and methodology information should be removed from this section ……………

Thanks for such remarks. Ok, this part was inserted in Materials and Methods, in its proper place under section 4.3. ( Antibiotic susceptibility test).

Supplementary Fig. 2A: How can be crude honey if it is diluted at different concentrations.

Ok, the stock honey bee was a native one from the farm. So, it was corrected to native honey bee and their dilutions.

Line 168-170: it must be rewritten. English grammer isnot corrected.

Ok, it was revised and corrected then written again and highlighted in the text.

Table 3 cant be thoroughly read. Please adjust it.

Ok, it was adjusted.

The resolution of figure 8 is not proper for the journal. It must be presented in the higher resolution.

Ok, The resolution of figure 8 is revised and is more obvious.

Line 236-241: the statement need to be reformulated……

Ok, the note is given and highlighted in the text.

Line 254 presents a contradiction. ……

Thanks  for this peer rigorous review. Ok, it was corrected and highlighted in the text. The strain was resistant to 10 antibiotics out of 12 ones tested.

Line 266: Again, as in my previous review, this expression has nonsense………………..

Ok, the sentence rewritten again. We did not mean further publication. The sentence in meaning of our students meant the rest and following experiments in this study. It is corrected and highlighted in the text.

Line 274-277: I previously asked for a better explanation………………..

Ok, the mechanism of action of the compounds elucidated by GC-MS analysis against bacterial cells are given and marked in the text. Since the goal of the work was not on mode of action, but it is necessary to be added to give full scientific knowledge on MSE-mechanism of antibacterial action of each chemical class was showed.

Line288: Why did not study the chemical characterization of the honey bee?

I respect the opinion of the reviewer. However, the natural agent giving the better antibacterial activity was the target of instrumental analysis under our standard conditions. It was difficult to do such GC-MS of honey bee (HB), because we are responsible for certain space of pages and lines for the manuscript. If the HB gave at our conditions, I mean against MDR S. aureus B3 strain, it was analysed instead of Moringa. It may give better antibacterial activity against other indicator bacterium.

Line 297-298: How the compounds of both extracts could act synergistically……………………..

We thank the reviewer for  this nice comment. We did not judge that a synergism occurred between MSE and tetracycline. Synergism may be occurred between both of them (not evident) due to binding of polar and non-polar chemical moieties of both of them and vice versa. This answer is given in details in the manuscript and marked by yellow colour.

Please correct across the manuscript honey bee not bee honey.

It was corrected in the whole manuscript.

Lines 441-443: How the dilution were made?

Native honey (100%) was dilutedby sterile distilled water. A 90% concentration for instance contain 90 mL honey bee + 10 mL distilled water. It is given in details in the manuscript and marked by yellow colour.

Lines 453-454 Please rewrite the sentence.

Ok, it is done.

Lines 475-476: it is not true that  all results were expressed as the mean ±  SD.

The statistical analysis (the mean ±  SD)  was done in Table 1 after reviewing again.

Reviewer 2 Report

Dear Authors,

Although you make some correction of your manuscript I still have some comments. Not all suggestions  were considered in the resubmitted version of your manuscript. For details please see the attached document. Comments are highlighted in blue color.

Please check your manuscript carefully in order to elimination of words and information repetition. In comments I give only examples, but whole manuscript must be corrected. I strong recommended language correction by native speaker.

Author Response

Reviewer 2# Round 2                                                                               

Please connect more strongly particular parts of manuscript, e.g. explain why oils from different plants were tested at the beginning of experiment?

Essential oils used in this article were obtained from El-Hawag factory, Bader city, Egypt, under the supervision of Ministry of Health license no: 150/80 for the year 2002, and they sterilized with 0.45 μm filter paper obtained from high lab company- Zagazig city-Sharkia. M.oleifera essential oil was mandatory to be used to compare between inhibition due to MSE and Such M.oleifera oils. Others two garlic and clove oils were used as they were available and as a comparable experiment. We were trying different natural agents that were available.

plants oil which will be more efficient in inhibition Maybe in order to choose this of investigated bacteria?

in results, as an This information should be added in manuscript in proper place, moringa oleifera was a subject for future investigation.

Ok. This information was added in the manuscript in results and pointed by yellow colour

Lines 132

Remove parentheses when Tables and Figures number are presented, Table 2 not Table (2). Similarly in lines 140, 146, 150, 366, 378.

The Parenthesis was removed within the manuscript

In received version of manuscript are not corrected both in case of tables (e.g. lines 495, or 578) and figures. Please again, check and correct.

All Table and Figure citation within the manuscript are corrected within the whole manuscript.

This information from ;line 467-478 are not necessary.

Not the record in Gene Bank but in you manuscript, italic or not?.

Thank you for this point. Ok, Gene Bank is italic.

Figure 5 and Figure 6 presents IZDs of S. aureus for 40 ug/mL of MSE, why the results are different? The differences between variants are statistically important? It is worth to consider this in discussion or results description

Thank you for these remarks in this point. The value of inhibition in Fig. 5 and.6 is an error from the statistical analysis. Perhaps, an error due to computation or personal error

I ask again, received results are statistically important? Please include this issue in the results.

Thank you, Ok. This issue added within  the manuscript.

Other comments (line numeration according to the highlighted version):

Line 23-33

“α-blood hemolysis; β-blood hemolysis”, next in line 125-126 “alpha and beta blood hemolysis” while in line 187 “α blood hemolysis, β blood”. Uniform in whole manuscript.

This error was corrected into α- and β-blood hemolysis” and mentioned within the whole manuscript by yellow colour.

Line 81,

Uniform bacteria names recording. Full name or abbreviation.

We thank the reviewer, this mistake was corrected

Line 100

What means SAC? Abbreviation could be used, after its explanation.

“Isolation and identification of SAC isolated from some Egyptian foods” – “isolation” and “isolated”, write in the other ways.

This error was repaired within the manuscript. SAC mean presumptive S.aureus colonies

The section title of (Isolation and identification of SAC isolated from some Egyptian foods) was rewritten into (Isolation and identification of presumptive S.aureus strains from some Egyptian foods)

Line 140 rewrite the section title.

Check the whole manuscript according proper spacing between words and numbers.

The section title was rewritten within the manuscript into (Molecular identification of the B3 strain by sequencing of 16S rRNA gene)

Proper spacing between words and numbers was checked.

Line 192

Repetition information from line 187, and 193. Please rewrite this fragment (line 186-194) in way avoiding repetition.

We thank the reviewer for this point; this fragment (line 186-194) was rewritten again and pointed by yellow colour with in the manuscript

Line 212

Why the common name of plants are writing in capital letter?

Sorry, this error was corrected, name of plants are writing in small letter within the whole manuscript.

Line 224

“Table 3” should be in parenthesis.

The parenthesis was added to Table 3

“Figure (5)” it is improper citation of figures, should be “Figure 5”. Similarly line 233. Check whole manuscript and correct the way of figures citation.

Figures citation were corrected within the whole manuscript and pointed by yellow colour

Line 358

Give the name of author in this way of citation: “ …..those received by Mama et al. [32] who…”

The way of this citation was corrected within the manuscript

Line 549-551

Please carefully check structure of the sentences and correct.

This error was corrected within the manuscript

Reviewer 3 Report

The authors have revised the manuscript as suggested.

Author Response

Reviewer 3# Round 2                                                                               

Comments and Suggestions for Authors

The authors have revised the manuscript as suggested.

Thank you very much for the reviewer.

Round 2

Reviewer 1 Report

TThe manuscript was improved, and almost all the suggestions were addressed. However, still, in the manuscript remain several English grammar mistakes. For example: lines 196, 197, 252, 310, 324, 381, 388, and so on...

Please, it is imperative to check every line, every letter, every expression to communicate the ideas accordingly. There are mistakes in the use of punctuation marks such as commas, semicolons, etc.

I'm afraid I have to disagree with the answer to the lack of honey bee characterization since it was used as an antimicrobial agent. It is always recommendable to characterize the substances used as antimicrobial agents to suggest the possible chemical interactions between components and the action. You can also cite some references to the composition if it is not the primary goal. However, limitation in the number of lines in a manuscript is not a sufficient reason to support an answer.

Also, check the space between paragraphs—for example, lines 194, 195, 196, but not limited to those.

The style of the tables is different. Is it possible to keep one type?

Figure 5 and figure 6 can have the same size, and figure 6 removes the legend that is not necessary since the name of the Y-axis is the same.

The best to improve figure 8 is to drop the data. Do not scan images since the resolution is too low. Use Origin or similar software to plot the data for FTIR.

Author Response

Replies to Reviewer # 1- Round 2                 

 Top of F

Comments and Suggestions for Authors

The manuscript was improved, and almost all the suggestions were addressed. However, still, in the manuscript remain several English grammar mistakes. For example: lines 196, 197, 252, 310, 324, 381, 388, and so on.

Thank you very much.  English grammar mistakes were corrected in the manuscript.

Line 196: 2.6. Antibacterial activity of some essential oils against S. aureus LC 554891 by disc diffusion assay

Please, it is imperative to check every line, every letter, every expression to communicate the ideas accordingly. There are mistakes in the use of punctuation marks such as commas, semicolons, etc.

Ok. Thank you very much. It was done.

I'm afraid I have to disagree with the answer to the lack of honey bee characterization since it was used as an antimicrobial agent. It is always recommendable to characterize the substances used as antimicrobial agents to suggest the possible chemical interactions between components and the action. You can also cite some references to the composition if it is not the primary goal. However, limitation in the number of lines in a manuscript is not a sufficient reason to support an answer.

 I thank the reviewer for carful revision. Of course the manuscript improved in its overall quality.  We were looking to analyze by GC-MS the bioactive compounds of the more active natural agent at our condition to be able to use it in combination with antibiotic. Honey bee was not the natural agent gave the more antibacterial activity.  The bioactive compounds of honey bee were described previously in details.

   I agree with the opinion of the reviewer to show in details the  bioactive compounds of honey bee and their mechanism of action against bacteria. Therefore, this last point was given and marked by green color in the text.

Honey bee has antibacterial nature due to presence of  H2O2, phenolic compounds and pH [10]. There are polyphenolic compounds present in honey bee  which  are responsible for its antibacterial activity . The common polyphenolic compounds are gallic acid, cinnamic acid, ferulic acid, hydroxyl cinnamic acid, sinapic acid, syringic acid and chlorogenic acid.  These compounds inhibit the bacteria by disrupt bacterial membrane, inhibit DNA gyrase, induce topaisomerase IV mediated DNA cleavage , inhibit peptidoglycan and ribosome synthesis [Victoria et al., 2019) 

Also, check the space between paragraphs—for example, lines 194, 195, 196, but not limited to those.

Thank you very much. The  space between paragraphs were adjusted.

The style of the tables is different. Is it possible to keep one type?

Ok. Thank you. The tables are one type now.

Figure 5 and figure 6 can have the same size, and figure 6 removes the legend that is not necessary since the name of the Y-axis is the same. 

Thank you very much for this point.  Figure 5 and figure 6 have the same size.  The legend is removed.

The best to improve figure 8 is to drop the data. Do not scan images since the resolution is too low. Use Origin or similar software to plot the data for FTIR.

Thank you very much. It was improved.

Reviewer 2 Report

Dear Authors,

Your manuscript is sufficient improved. I have only a minor, editorial comments, please see below..

Lines 110-111, 114,

This is misunderstanding, I used quotation marks when cited the original fragment of your manuscript and when gave the proper description of tables. So please, in lines 110-111 and 114 instead of quotation marks use parentheses.

Lines 185, 187, 213, 252 and 594

Remove inverted commas (e.g. not “Figure 4A” but Figure 4A )

Lines 239, 244

Figure 5 and 6, please add information, according to methods section: Mean value with the standard error.

Line 365

Not Mama [37] but Mama et al. [37]

Author Response

Replies to Reviewer # 2- Round 2        

General remarks

Comments and Suggestions for Authors

Dear Authors,

Your manuscript is sufficient improved. I have only a minor, editorial comments, please see below..

Lines 110-111, 114,

This is misunderstanding, I used quotation marks when cited the original fragment of your manuscript and when gave the proper description of tables. So please, in lines 110-111 and 114 instead of quotation marks use parentheses.

Thank you very much. We use parentheses  in lines 110-111 and 114 in the manuscript. The corrected parts highlight by green colour within the text.

respectively within the positive samples that showed bacterial counts (Supplementary Table 1). T

belonging to S. aureus bacterium (Supplementary Table 2).

Lines 185, 187, 213, 252 and 594

Remove inverted commas (e.g. not “Figure 4A” but Figure 4A)      

Thank you. We removed the inverted commas in the text.

Lines 239, 244

Figure 5 and 6, please add information, according to methods section: Mean value with the standard error.

Thank you. Mean value with the standard deviation was added to Fig. 5 and 6.  All values reflect the mean values of 3 replicates and standard deviations.

Line 365

Not Mama [37] but Mama et al. [37]

Thank you very much. It was corrected in the text.

with Mama  et al. [37]

This manuscript is a resubmission of an earlier submission. The following is a list of the peer review reports and author responses from that submission.

Round 1

Reviewer 1 Report

The manuscript deals with an interesting topic about the uses of natural extracts to control S. aureus isolates from different food sources. However, from a previous review of the paper, there were several points that I requested to improve, and those were not assessed. For example, mistakes in the written (typos and miss of sense) in lines 21, 22, 24, 31, 60, 121, 128, 133, 134, 243, etc., but not limited to them. Also, I requested to add more information about Moringa in the introduction, but that was also not assessed. Surprisingly, the conclusions are too short and do not highlight the findings of the work. That, too, was highlighted previously. 

In the introduction section, lines 57-58, it is better to move that objective to the end of the introduction section.

The introduction section needs a paragraph were the uses of M. oleifera extracts are demonstrated as antibacterial agents but also, what other studies have been performed with natural extracts as S. aureus MRD inhibitors for comparison purposes even in the discussion section.

The final paragraph with the goal of the work needs to be rewritten to be more understandable.

Add more information about M. oleifera.

In the introduction, it is not explained anything related to the uses of bee honey (BH) as antimicrobial. Please, add some information about it.

I suggest adding more discussion about the antimicrobial mechanism observed based on the chemical characterization by CG-MS in work.

Where is the FTIR graph that is claimed in the results section? (lines 165-168)

I strongly suggest reviewing and elaborate on the conclusions. Several results need to be highlighted according to work.

Since it is a resubmission, I recommend a point by point answer to each request by the reviewers. 

Reviewer 2 Report

Dear Authors,
I have some suggestion how improve your manuscript. Please see comments.

General comments:
Literature in the literature section should be listed in the order accordance to the citation in the text, please check (e.g. lines 51, 222).
Writing style and language need to be improve.
All abbreviation should be explain before first use in the text.
Check the use of capital letters in text and correct when neceseary, there are many words that should be written in lowercase (lines 136, 26,etc.).
Please connect more strongly particular parts of manuscript, e.g. explain why oils from different plants were tested at the beginning of experiment?

Particular comments:

Lines 2-4

Title. Moringa oleifera seed extract was combine either singly or only with tetracycline, one antibiotic, not antibiotics.

Line 60

Add a space after 40% and in lines 329, 331 etc. Check whole MS for spacing.

Lines 67-70, 73-77, 266,

Please check punctuations. Instead of semicolons use rather commas.

Line 76

“Moringa oleifera seed extract (MSE) was used as an inhibitory agent ...” rather was tested as an inhibitory agent.

Line 80

Not “n 25, n 50” but n=25 and n=50, please unified.

Line 84

“by about 45.45% and 2%” Please check this data with Supplementary Table 1. 

Line 88-89

“The 18 S. aureus strains obtained showed positive results regarding coagulase reaction, α and β –blood hemolysis.”

All 18 strains or only S. aureus B3 strain as is written in section 2.5?

Proper place for this information is section 2.5.

Line 109

Do you mean Supplementary Figure 1, please check.

Line 114

Please check the designation.

Lines 132

Remove parentheses when Tables and Figures number are presented, Table 2 not Table (2). Similarly in lines 140, 146, 150, 366, 378.

Line 139-140

“The inhibitory effect of MLE, MSE and BH against S. aureus LC 554891 is given in Table (3) and Supplementary Figure (1).” or Figure 2, please check.

Line 140

What means IZDs? It should be clearly explain before first use in the text. Similarly line 142, EE?

Line 175

“18% of foods”. I suggest to add - tested foods.

Line 188

“The 18 presumptive S. aureus bacteria (SAC)....” rather bacterial strains. Similarly line 192

Lines 243, 245

What means “CSME”?

Line 263

“Potato chips” – use lowercase letter. Check the whole text for correctness of writing upper and lowercase letters.

Line 267, 361

“Km” or rather km?

Lines 268-270

Check the correctness of the sentence, avoid word repetition, abbreviation can be used after previous explanation.

Lines 273-279

Add source literature.

Line 314-315

“...volume tubes according to [44].” Rather “...volume tubes according to Williams et al. [44].

Line 315

Add space after dot. Check whole manuscript in this regard.

Line 330

Check the way of gene record.

Lines 337

Check the spelling Latin names

Line 362-363

In which way different concentration of honey (20-90%) were received? Please add this information. In Table 3, 50 and 100% variants are presented, add short explanation why?

Line 367

It was possible to use automatic pipette for high concentrated honey application?

Line 372

“Serial dilutions of MSE were made in sterile deionized water at a ratio of 1:1 (v/v).” in this way received concentration from 10 to 70 ug/mL? Please explain.

Line 394

"...plus the standard error”, use rather "with the standard error".

Table 2. Sources of extracts are not mentioned in the text (moringa oil, garlic oil, clove oil). Additionally please use unify way of description this substances, compare with line 337.

Table 3 presents results inhibition of bee honey at concentrations and 100%. It means that natural, crude honey were used?

For MLE and MSE the concentration was in ug/mL? In “Supplementary Figure 2B. Antibacterial activity of the initial aqueous extract (10g/100mL water)” - There was realy so high concentration? And next were reduced to ug/mL as is shown in Table 3?

Figure 6 and Table 3 presents IZDs of S. aureus for tetracycline in concentration 10µg, why in Table 3 is given value equal 0 and in figure 6 is higher than 0?

Figure 5 and Figure 6 presents IZDs of S. aureus for 40 ug/mL of MSE, why the results are different? The differences between variants are statistically important? It is worth to consider this in discussion or results description.

Line 430

Section “References”, please use unified way of literature record, especially the numbers of pages.

Figure 1. Use capital letter when start the sentence.

Reviewer 3 Report

In this study, Enan et al isolated a MDR S. aureus strain LC554891 and evaluated the activity of Moringa oleifera extracts against LC554891. I have some comments described as follows,

1. The author isolated 18 S. aureus strains from beef luncheon, potato chips, and corn flakes (Table 1). However, the number of isolate is limited to study the prevalence of antimicrobial resistance of S. aureus in Egypt.

2. The authors determined the inhibitory activity of both Moringa oleifera extracts (MLE/MSE) and bee honey (BH) against S. aureus (Table 2). Although the results showed antibacterial activity of essential oils against S. aureus LC554891, the authors should test more S. aureus strains. 

3. The authors performed GC-MS analysis on M. oleifera seed extracts and identified putative bioactive compounds. However, the authors should purify these compounds to characterize their antibacterial activity against S. aureus.